# Comparison of Representation Learning Techniques for Tracking in time resolved 3D Ultrasound

**Daniel Wulff** [1]                                                   WULFF@ROB.UNI-LUEBECK.DE
**Jannis Hagenah** [1]                                          HAGENAH@ROB.UNI-LUEBECK.DE
**Floris Ernst** [1]                                                     ERNST@ROB.UNI-LUEBECK.DE

[1] *Institute for Robotics and Cognitive Systems, Universität zu Lübeck, Lübeck, Germany*

## Abstract

3D ultrasound (3DUS) becomes more interesting for target tracking in radiation therapy due to its capability to provide volumetric images in real-time without using ionizing radiation. It is potentially usable for tracking without using fiducials. For this, a method for learning meaningful representations would be useful to recognize anatomical structures in different time frames in representation space (r-space). In this study, 3DUS patches are reduced into a 128-dimensional r-space using conventional autoencoder, variational autoencoder and sliced-wasserstein autoencoder. In the r-space, the capability of separating different ultrasound patches as well as recognizing similar patches is investigated and compared based on a dataset of liver images. Two metrics to evaluate the tracking capability in the r-space are proposed. It is shown that ultrasound patches with different anatomical structures can be distinguished and sets of similar patches can be clustered in r-space. The results indicate that the investigated autoencoders have different levels of usability for target tracking in 3DUS.

**Keywords:** radiotherapy, deep learning, latent space, clustering

## 1. Introduction

Ultrasound imaging is frequently used in clinics due to its capability to visualize soft tissue in real-time without damaging radiation. Furthermore, 3D Ultrasound (3DUS) is a promising modality for tracking in radiation therapy (Ipsen et al., 2019, 2021). A first approach to learn representations of 3DUS patches using a conventional autoencoder was proposed in (Wulff et al., 2020). With that, comparing patches can be performed by measuring distances in the representation space (r-space). However, an investigation for representation learning techniques that potentially perform better for a tracking task is still missing. To perform tracking in the r-space the allocation of patches in the r-space has to be meaningful in terms of differentiation of patches containing similar or dissimilar structures. In this study, three different r-spaces created by different autoencoders are used to investigate their usability for tracking. By clustering sets of deformed and translated patches generated to simulate temporal and spatial shifts, respectively, the usability of using the r-spaces for tracking is analyzed using two different metrics to compare the cluster results and the ground truth.

## 2. Methods and Results

The r-spaces of three different kinds of autoencoders (AEs) are analyzed: 1. Conventional AE (cAE), 2. Variational AE (VAE) and 3. Sliced-Wasserstein AE (SWAE) (Kolouri et al., 2018). The AEs are trained using patches of real liver ultrasound data of one subject from the dataset provided in (Ipsen et al., 2021) and the model architecture presented in (Wulff et al., 2020). However, the different AEs differ in its specific attributes. cAE directly maps a patch into a 128-dimensional representation vector, VAE maps a patch into a 128-dimensional normal distribution and SWAE maps the patches into a 128-dimensional hyperbullet shaped r-space. The expected motion of an anatomical structure can be separated in translation, rotation and deformation. For simplification rotation is neglected here. Translation means that the target is shifted within the patch so there should exist another patch in the US volume in which the target position fits better to the reference. Deformation means that the target position within the patch is correct, but the target shape changed. This motion is expected between different time frames and means that no even more similar patch exists in the US volume as the target location is correct. Regarding tracking, deformed patches should be close and translated patches should be apart in the r-space as the current target location is searched. For the experiments, ten cubic patches with a side length of 30 voxels are selected from different time frames at random locations. These patches are augmented artificially to generate realistic temporal and spatial shifted patches. For each patch a subset of 50 deformed patches is generated randomly using *elasticdeform* library where a grid of size $5^3$ is used and the deformation magnitude is sampled from a normal distribution with $\sigma = 1$ (van Tulder, 2021). In addition, a subset of 50 translated patches is generated by shifting the patch location in the ultrasound volume randomly in a range of $\pm 10$ voxels in all directions to be sure that the target stays visible in the patch. These two sets of patches are encoded using the encoder part of the AEs to get the r-spaces. To evaluate if the subsets are separable k-means cluster algorithm is performed to the r-space sets. The results are compared to the known ground truth and the rate of correct clustered representations (precision) is determined to quantify the results. In addition, the Calinski-Harabasz (CH) coefficient, is used to quantify the ratio between the internal dispersion of a subset and the dispersion between the subsets (Calinksi and Harabasz, 1974). The higher the coefficient, the better is the ratio. The results of both experiments are given in Table 1.

Table 1: Precision and CH score of clustering results in representation space of conventional AE (cAE), Variational AE (VAE) and Sliced-Wasserstein AE (SWAE)

| Data Type | Autoencoder | Precision | CH score |
|---|---|---|---|
| Deformation / Translation | cAE | 1.0 / 0.6 | 1197 / 28 |
| | VAE | 0.8 / 0.5 | 60 / 8 |
| | SWAE | 1.0 / 0.7 | 2385 / 33 |

## 3. Discussion and Conclusion

Considering the results, all tested AEs are able to represent the patches, but clustering them by their representations was less effective in the translation experiment. This indicates that the location of the structure inside the patch is coded in the representation. Patches with a translated target have a high distance in the r-space so the subsets can not be separated. This aspect is confirmed by the small CH scores compared to the deformation experiment. Concerning the aim of performing tracking, this is a promising characteristic. In the deformation experiment the precision is even 1.0 in cAE and SWAE so the clusters can be separated without errors. In the VAE r-space the precision is the lowest in both experiments. This can be caused by the sampling procedure from a normal distribution in the encoder part of the VAE. In the deformation experiment the CH scores of both cAE and SWAE are high. This indicates that the patches inside a subset are close as well as the distances between the subsets are high. For performing tracking, this is advantageous as patches showing the same structure from different time frames are close and patches showing different structures are far away from each other. In this study, a method to analyze r-spaces for their usability for tracking is proposed. This study indicates that SWAEs and cAEs are more suitable for tracking tasks than VAEs due to higher separation performance of a set of dissimilar patches by its representations. Due to the very high CH score that is achieved in the SWAE r-space, this AE seems to be a promising technique for tracking in time resolved 3DUS. In future studies tracking in representation space of a SWAE will be performed.

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
