# OpenReview forum: "Comparison of Representation Learning Techniques for Tracking in time resolved 3D Ultrasound"
_MIDL.io/2021/Conference/Short — MIDL 2021 Poster_

### Official Review · Reviewer_B7ji · 2021-04-21

**Confidence:** 4
**Final Rating:** 3

**Summary:**

This paper presents an analysis of the latent space produced by different autoencoder architectures for tracking applications in 3D ultrasound images. In particular, the space generated by conventional, variational, and sliced-wasserstein autoencoders is compared by means of the clusters computed over their feature representation. It is demonstrated that the different architectures separate the clusters in different ways, and hence that they offer different expressive power in the separation of the patches.

**Strengths:**

- **Motivation** The study of different autoencoder representations for 3DUS image patches is interesting. Moreover, the paper analyses feature learning in cases of translation and deformation of targets which are relevant scenarios.
- **Outcomes** The potential outcomes of this work could improve the understanding of representation learning for 3DUS tracking (and potentially other) applications. Moreover, this analysis could lead to enhanced unsupervised feature representation learning, reducing the need of labeled data.

**Weaknesses:**

- **Impact on Tracking** Even though the conclusions of this paper are clear from the analysis of the latent spaces, the authors show just that the feature representations can be separated with different outcomes depending on the architecture. Particularly, it is not demonstrated how the different representations impact on the tracking performance. It would be interesting to have some quantitative results in Table 1 (and relative discussion) about how the performance of a 3DUS tracker changes while employing the features of the three different autoencoders.
- **Presentation** There are some sentences that are not clear along the paper that need to be rephrased. E.g. in the abstract the sentence "For this, a method for learning meaningful representations with which recognizing anatomical structures in different time frames is capable would be useful." is not clear. "For performing tracking" -> "To perform tracking". "This motion is expected between different time frames and means no even more similar patch exists." not clear. "three dierent r-spaces created by different autoencoders are used to investigate its usability for tracking." "its usability" -> "their usability".

**Deanonymize Review:**

no

**Detailed Comments:**

I suggest the authors to add the performance of a tracker that uses the different representations to make their analysis stronger.

**Justification Of The Rating:**

The paper gives interesting and potentially useful insights on the representation learning of 3DUS image patches through different autoencoder architectures. There are some issues with the presentation but, if addressed, do not prevent the publication of the paper.

**Paper Type:**

validation/application paper

**Special Issue:**

no

---

### Official Review · Reviewer_cir4 · 2021-04-28

**Confidence:** 4
**Final Rating:** 2

**Summary:**

The authors propose a comparison of how good are three different learned representations of 3D ultrasound patches for patch matching, towards a future implementation in a tissue tracking algorithm. Three autoencoder models are compared, using ability to cluster patches by translation an deformation (with respect to a ground truth) using k-means on the representation space. The results suggest that all models produce a representation that can be better clustered deformation than to cluster translation.

While I agree that using learned representations for tracking is promising, I have concerns about the conclusions that authors draw from their results: either I have misunderstood what the experiments do, in which case I would suggest to improve the clarity of the paper; or I have understood it correctly and the conclusions are wrong.

**Strengths:**

* Using representation learning to encode patches for tracking is an interesting idea, which can potentially alleviate the need to craft features for ultrasound tracking
* Comparing different models can help towards choosing the best representation strategy

**Weaknesses:**

* Only synthetic data was used. There is publicly available (for example, the MICCAI Clust challenge - https://clust.ethz.ch/) which would have been more interesting and realistic to use.
* The work seems to be in a very early stage -no applicable for tracking yet.
* Intepretation of what the results mean may be wrong -please check my detailed comments.
* Experiment set up (range of synthetic translation) is not well justified.

**Deanonymize Review:**

yes

**Detailed Comments:**

Authors interpret the low accuracy of the representations in the translation experiment as "the location of the structure in the patch has high impact in the representation so the subsets can not be separated.". However I think that the interpretation is the opposite: the location of the structure in the patch does not have a high impact in the representation, hence the representations cannot be clustered correctly. In other words, the representation is fairly insensitive to the location of the structure. Moreover, this makes sense because of the pooling strategy in the encoder, which is known to introduce (some) translation invariance. With the little detail provided about the architecture it is not possible to verify this.

Depending on the tracking application, translation may be more important than deformation; for example, features may be locally not deformed, but they may move for instance from probe motion or from respiratory/organ motion. As a result, I would argue that ensuring that translation works is more important than having deformation right. Also, the choice of a a translation range of +-10 voxels seems arbitrary and is not linked to the expected motion for a given application or the image resolution, so it is difficult to tell how useful that is.

**Justification Of The Rating:**

The work seems to be in a too early stage for publication. The experiments have been carried out exclusively on synthetic data, although there is real data publicly available. Moreover, the experimental set up, and the range of translations applied is not justified.  The interpretation of the obtained results may be incorrect.

**Paper Type:**

methodological development

**Special Issue:**

no

---

### Meta-Review · Program_Chairs · 2021-05-10

**Recommendation:** Accept (Poster)
**Confidence:** 4

**Metareview:**

The paper presents a comparison of different representation learning approaches and their usefulness for tracking in US images. While the study is in an early stage, as reviewers point out, it already gives some useful insights. I think the main critique from the more critical reviewer stems from a misunderstanding of what is clustered and why - which is indeed unclear in the text. Authors should clarify these points in the final version and consider strengthening the experiments on translation (absence of clustering dose not yet prove usefulness for tracking).

---

### Decision · Program_Chairs · 2021-05-11

Accept (Poster)